# Differentiable and Learnable Wireless Simulation with Geometric Transformers

**Thomas Hehn, Markus Peschl, Tribhuvanesh Orekondy, Arash Behboodi, Johann Brehmer**[*]
Qualcomm AI Research[†]

## Abstract

Modelling the propagation of electromagnetic wireless signals is critical for designing modern communication systems. Wireless ray tracing simulators model signal propagation based on the 3D geometry and other scene parameters, but their accuracy is fundamentally limited by underlying modelling assumptions and correctness of parameters. In this work, we introduce Wi-GATr, a fully-learnable neural simulation surrogate designed to predict the channel observations based on scene primitives (e. g., surface mesh, antenna position and orientation). Recognizing the inherently geometric nature of these primitives, Wi-GATr leverages an equivariant Geometric Algebra Transformer that operates on a tokenizer specifically tailored for wireless simulation. We evaluate our approach on a range of tasks (i. e., signal strength and delay spread prediction, receiver localization, and geometry reconstruction) and find that Wi-GATr is accurate, fast, sample-efficient, and robust to symmetry-induced transformations. Remarkably, we find our results also translate well to the real world: Wi-GATr demonstrates more than 35% lower error than hybrid techniques, and 70% lower error than a calibrated wireless tracer.

## 1 Introduction

Modern communication is wireless: more and more, we communicate via electromagnetic waves through the antennas of various devices, leading to progress in and adoption of mobile phones, automotive, AR/VR, and IoT technologies (Chen et al., 2021; Dahlman et al., 2020). All these innovations build upon electromagnetic (EM) wave propagation. Therefore, modelling and understanding wave propagation in space is a core research area in wireless communication, and remains crucial as we are moving toward new generations of more efficient and spatially-aware wireless technologies.

How can one model the influence of spatial environments on EM waves transmitted by one antenna and received by another? *Model-based* approaches (specifically wireless ray tracing methods) are popularly used, which identify physical propagation paths and their corresponding characteristics (e. g., power, phase) between the two antennas. The propagation paths are naturally influenced by the environment, causing phenomena such as reflection, diffraction, and transmission. Although popular, model-based approaches have shortcomings: Due to their approximate nature, they suffer from a sim-to-real gap in non-trivial environments, and in most cases they are neither learnable nor differentiable.

*Hybrid* and *fully-learnt* approaches attempt to address some shortcomings of model-based methods. *Hybrid* approaches (Orekondy et al., 2022b; Hoydis et al., 2023) retain some aspects of model-based techniques (e. g., evaluating physical propagation paths), and introduce learnable components (e. g., neural evaluation of materials). The learnable parameters help with incorporating real measurement data to reduce the sim-to-real gap. However, similar to model-based approaches, they are once again bottle-necked by availability of accurate and efficient implementations of explicit physical models. In contrast, *fully-learnt* approaches (Hehn et al., 2023; Levie et al., 2021; Lee & Molisch, 2024), propose an end-to-end neural surrogate that consumes a visual representation of the scene and predicts spatial wireless characteristics. The core idea, which we too leverage in this work, is to enable the neural surrogate to implicitly learn the physical models that best explain the

---

[*]Work was completed while an employee at Qualcomm Technologies Netherlands B.V.
[†]Qualcomm AI Research is an initiative of Qualcomm Technologies, Inc.

observations. While there have been some advances in this direction, we find the choice of architectures ill-suited and inefficient for the task.

Existing *fully-learnt* surrogates largely rely on CNN-based architectures and are limited to lossy 2D representations of the scene (i. e., top-down binarized images). We argue that wireless propagation is inherently a three-dimensional *geometric problem*: a directional signal is transmitted by an oriented transmitting antenna, the signal interacts with oriented surfaces in the environment, and the signal eventually impinges an oriented receiving antenna. For this reason, it is critical for neural surrogates to model and flexibly represent 3D geometric aspects in the propagation environment.

Learning neural surrogates directly on 3D data presents two significant challenges. The first challenge involves representing the diverse 3D geometric input data such that a neural network can efficiently learn the signal propagation interactions among scene elements. The second challenge is ensuring that the neural network can generalize to unseen scenes, given that the received wireless signal remains invariant under rotations, translations, and reflections of the entire scene.

In this work, we aim to tackle these challenges with a novel fully-learnt approach to model wireless signal propagation in 3D. We propose Wi-GATr, a backbone based on Geometric Algebra Transformers (GATr) (Brehmer et al., 2023) for surrogate models utilizing 3D geometric representations and strong geometric inductive biases. A key component is a new tokenizer that defines the input representation for the diverse, geometric 3D data of wireless scenes. Furthermore, this architecture is $E(3)$-equivariant with respect to the symmetries of wireless propagation, but maintains the scalability of a transformer architecture.

A geometric, fully-learnt treatment of modelling the influence of simulation parameters (e. g., scene mesh, antenna position, orientation) on resulting wireless measurements (e. g., receive signal strength) offers many advantages. First, the underlying inductive biases of Wi-GATr enable sample-efficient learning with the resulting predictions being robust to certain variations, such as the choice of coordinate frame. Second, since Wi-GATr is fully differentiable w.r.t. all its simulation parameters, we can leverage it to tackle inverse problems (e.g., localization). Finally, Wi-GATr can be easily adapted into a generative framework and thereby allowing probabilistic inferences.

We present a comprehensive evaluation of Wi-GATr across various tasks and use cases. To facilitate large-scale evaluation and training, we introduce two novel wireless datasets, `Wi3R` and `WiPTR`, each comprising thousands of indoor scenes with varying complexity. Our results show that Wi-GATr outperforms competing approaches. For instance, on `WiPTR`, Wi-GATr achieves an MAE of 0.64 dB using only 10% of the training dataset, surpassing PLViT (1.28 dB) and a Transformer (0.69 dB) trained on the full data. Furthermore, we demonstrate the versatility of our surrogate by leveraging the differentiability of Wi-GATr for receiver localization, achieving accuracy up to 60 cm. Additionally, we show how Wi-GATr can serve as a generative diffusion model trained on joint simulation parameters, enabling the reconstruction of various scene variables, including the intricate mesh geometries. We also train and evaluate Wi-GATr on the real-world measurement dataset DICHA-SUS (Euchner et al., 2021), where it achieves state-of-the-art results. Specifically, Wi-GATr outperforms both hybrid approaches (with >35% reduction in error) and a calibrated wireless ray tracer (with >70% reduction in error). Overall, our results indicate that our proposed approach Wi-GATr takes a promising step towards a sample-efficient and robust fully-learnt simulator that can effectively leverage the underlying 3D scene parameters to simulate wireless propagation effects.[1]

## 2 BACKGROUND AND RELATED WORK

**Wireless signal propagation.** How do wireless signals propagate from a transmitting antenna (Tx) to a receiver antenna (Rx) in a (static) 3D environment? While the system is fundamentally described by Maxwell's equations, for many realistic problems the ray approximation of geometric optics suffices (Keller, 1962). It approximates the solution to Maxwell's equations as a sum of planar waves propagating in all directions from Tx. Each planar wave is represented as a ray, characterized by various attributes (e. g., power, phase, delay) since transmission. As a ray reaches an object—that is, it intersects with its mesh—the interaction is modelled as reflection, refraction, or diffraction. During such interactions, the power, phase, polarization, and propagation direction of the wave can

---

[1]Our Wi-GATr code is available at `https://github.com/Qualcomm-AI-research/Wi-GATr`.

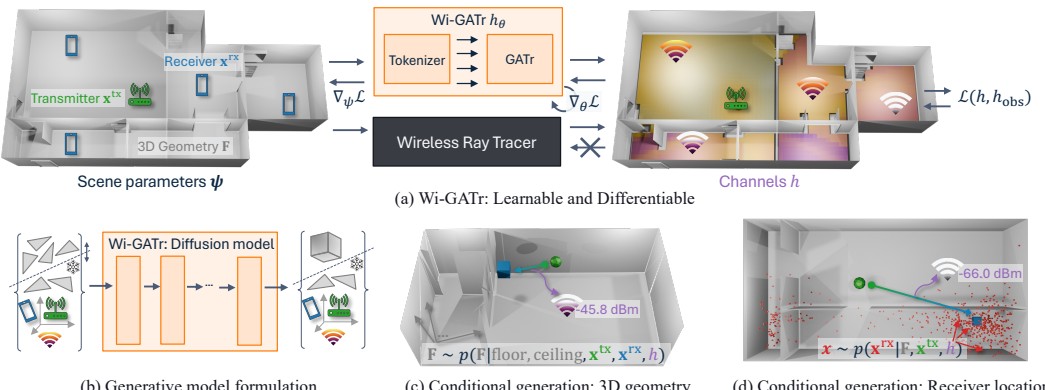

(a) Wi-GATr: Learnable and Differentiable

(b) Generative model formulation     (c) Conditional generation: 3D geometry     (d) Conditional generation: Receiver location

**Figure 1: Wi-GATr as a geometric neural surrogate for wireless simulation. (a)**: Akin to a ray tracer, Wi-GATr as a forward-model maps the 3D geometry, transmitter, and receiver properties to desired channel information, such as the received power. Wi-GATr is differentiable and can provide useful gradients with respect to its inputs. **(b)**: In a probabilistic approach based on diffusion, Wi-GATr can reconstruct 3D environments **(c)** and antenna positions **(d)** from the wireless signal.

change in complex, material-dependent ways. In addition, new rays can emanate from the point of interaction. After multiple interactions, the rays eventually reach the receiving antenna. The Tx and Rx are then linked by a connected path $p$ of multiple rays. The effects on the received signal are described by the channel impulse response (CIR) $h(\tau) = \sum_p a_p \delta(\tau - \tau_p)$, where $a_p \in \mathbb{C}$ is the complex gain and $\tau_p$ the delay of the paths at the receiver (Tse & Viswanath, 2005).

**Wireless channel models and simulation.** Wireless propagation models play a key role in design and evaluation of communication systems, for instance by characterizing the gain of competitive designs in *realistic* settings or by network performance as in base station placement for maximal coverage. Statistical approaches (3GPP TR 38.901) represent propagation as a generative model where the parameters of a probabilistic model are fitted to measurements. On the other hand, wireless ray-tracing approaches (Remcom; Amiot et al., 2013; Hoydis et al., 2022) approximate wave propagation using geometric optics principles: propagation paths between a transmit and receive antenna are estimated, and corresponding per-path characteristics (e. g., time-of-flight) are calculated. They are sufficiently accurate in many use cases and do not require expensive field measurement collection campaigns.

**Neural wireless simulations.** Both statistical and ray-tracing simulation techniques are accompanied by their own shortcomings, subsequently mitigated by their neural counterparts. Neural surrogates for statistical models (Ye et al., 2018; O'Shea et al., 2019; Dörner et al., 2020; Orekondy et al., 2022a) reduce the amount and cost of measurements required. Neural ray tracers (Orekondy et al., 2022b; Hoydis et al., 2023; Zhao et al., 2023) address the non-differentiability of simulators using a NeRF-like strategy (Mildenhall et al., 2020) by parameterizing the scene using a spatial MLP and rendering wireless signals using classic ray-tracing or volumetric techniques. While these techniques can be faster than professional ray tracers, they are similarly bottlenecked by expensive bookkeeping and rendering steps (involving thousands of forward passes). In contrast, we propose a framework to simulate wireless signals with a single forward pass through a geometric transformer that is both sample-efficient and generalizes to novel scenes.

**Geometric Algebra Transformer.** The growing field of geometric deep learning (Bronstein et al., 2021) aims to incorporate structural properties of a problem into neural network architectures and algorithms. A central concept is *equivariance* to symmetry groups (Cohen, 2021): a network $f(x)$ is equivariant with respect to a group $G$ if its outputs transform consistently with any symmetry transformation $g \in G$ of the inputs, $f(g \cdot x) = g \cdot f(x)$, where $\cdot$ denotes the group action. The Geometric Algebra Transformer (GATr) (Brehmer et al., 2023) is an E(3)-equivariant architecture for geometric problems, where E(3) is the three dimensional Euclidean group. We will show that this representation is particularly well-suited for wireless channel modelling. Second, GATr is a transformer architecture (Vaswani et al., 2017). It computes the interactions between multiple tokens through scaled dot-product attention. With efficient backends like FlashAttention (Dao et al., 2022),

the architecture is scalable to large systems, without any restrictions on the sparsity of interactions like in message-passing networks.

## 3    THE WIRELESS GEOMETRIC ALGEBRA TRANSFORMER (WI-GATR)

Our goal is to model the wireless channel, i.e., the interplay between the transmitted wireless signal, the 3D environment, and the transmitting and receiving antennas. We first outline the specific problem setup before we provide the details of how we tokenize the environment in geometric algebra primitives that provide the foundation of our Wi-GATr backbone.

### 3.1    PROBLEM SETUP

**Wireless simulation: fully-learnt forward model.**   The fundamental problem in simulating wireless propagation is to model an accurate *forward* process: mapping simulation parameters $\psi$ to the 'wireless channel' $h$ which describes the transformation between the transmitted signal and the received signal. In this work, we propose a fully-learnable surrogate $h_\theta(\psi)$ for the forward process. The core idea is to implicitly learn physical propagation models that best explain the observations.

**Simulation inputs: parameters $\psi$.**   A simulation scenario is defined by a large set of parameters $\psi$. Similar to prior works, we consider three groups of representative parameters: (a) *scene geometry and materials $F$:* parameterized as a 3D mesh, with each face associated with a discrete material class; (b) *transmit antenna properties $x^{tx}$:* parameterized by 3D position and orientation; (c) *receive antenna properties $x^{rx}$:* also parameterized by 3D position and orientation. Note that in our setting, we assume that antenna radiation patterns are shared between all receivers and transmitters, respectively, and as a consequence, they need to be learned implicitly. To the best of our knowledge, we present the first fully-learnt surrogate that works on the full 3D geometry (as opposed to 2D/2.5D image representations).

**Simulation outputs: wireless channel $h$.**   For a fixed set of simulation inputs, we want the simulator to predict the wireless channel $h$. Since we propose a learnable model, we rely on channel measurements for supervised training which can be collected using conventional devices (e.g., mobile phones). Measuring the exact channel $h$ is challenging (Euchner et al., 2021), which is why we describe the signal transformation by averaged statistics of the channel. Similar to prior works, we learn to predict real-valued, scalar channel characteristics $h$: non-coherent received power (Section 5.1), band-limited received power, and band-limited delay spread (both Section 5.4).

### 3.2    WI-GATR BACKBONE

We want a learnable forward-model for simulations $h_\theta(\psi)$, that can effectively leverage a wireless scene $\psi$ to reason about spatially-varying channels $h$ (e.g., receive power). Our core insight is that the forward-process is inherently *geometric*: electromagnetic waves are emitted by an oriented transmit antenna, interact (e.g., reflect and refract) multiple times with various surfaces in space and finally reach an oriented receive antenna at a different location. To exploit this inherent geometric nature of propagation, we propose 'Wi-GATr', a Wireless Geometric Algebra Transformer (Wi-GATr) backbone. Wi-GATr consists of two main components: a tokenizer and a network architecture. The tokenizer embeds the information of the wireless scene into geometric algebra while the network learns to model the channel. We now elaborate on these individual components.

**Wireless GA tokenizer.**   The tokenizer takes as input information characterizing simulation parameters $\psi$ and outputs a sequence of tokens that can be processed by the network. A key challenge of the tokenizer is efficiently representing a variable set of diverse data types. In our case, the data types range from 3D environment mesh $F$, which features three-dimensional objects with dielectric material properties such as buildings, to antennas $x^{tx}$ and $x^{rx}$ characterized through a point-like position, an antenna orientation, and additional information about the antenna type, and the characteristics of the channel $h$.

To support all of these data types, we propose a new tokenizer that outputs a sequence of geometric algebra (GA) tokens. Each token consists of a number of elements (channels) of the projective geometric algebra $\mathbb{G}_{3,0,1}$ in addition to the usual unstructured scalar channels. We define the GA

precisely in Appendix A. Its main characteristics are that each element is a 16-dimensional vector and can represent various geometric primitives: 3D points including an absolute position, lines, planes, and so on. The richly structured space of projective geometric algebra $\mathbb{G}_{3,0,1}$ is thus ideally suited to represent the different elements encountered in a wireless problem. It naturally supports scalars for wireless scalar fields (e. g., power, delay spread), points for positions (e. g., antennas, mesh vertices), vectors for orientations (e. g., antennas), and planes for surfaces (e. g., mesh faces). Details of our tokenization scheme are specified in Appenndix B.

**Network.** After tokenizing, we process the input data with a Geometric Algebra Transformer (GATr) (Brehmer et al., 2023). This architecture naturally operates on our $\mathbb{G}_{3,0,1}$ parameterization of the scene. It is equivariant with respect to permutations of the input tokens as well as E(3), the symmetry group of translations, rotations, and reflections. These are exactly the symmetries of wireless signal propagation, with one addition: wireless signals have an additional reciprocity symmetry that specifies that the signal is invariant under an role exchange between transmitter and receiver. We will later show how we can incentivize this additional symmetry property through data augmentation.[2] Finally, because GATr is a transformer, it can process sequences of variable lengths and scales well to systems with many tokens. Both properties are crucial for complex wireless scenes, which can in particular involve a larger number of mesh faces.

## 3.3 Applications beyond forward modelling

In the previous section, we proposed a Wi-GATr backbone for forward-simulations $h_\theta(\psi)$. We now highlight two additional benefits of the proposed backbone. First, exploiting differentiability of $h_\theta$ to solve inverse problems. Second, adapting the Wi-GATr within a generative model formulation and thereby allowing probabilistic inferences (either for forward, or inverse problems).

**Inverse problems solved by gradient descent.** The differentiability of surrogate models make them well-suited to solve inverse problems. More generally, the differentiability enables gradient-based optimization of input data (i. e., simulation parameters) to minimize a given loss. For instance, we can use the surrogates for receiver localization. Given a 3D environment $\boldsymbol{F}$, transmitters $\{\boldsymbol{x}_i^{\text{tx}}\}$, and corresponding signals $\{h_i\}$, and a random initialization $\boldsymbol{x}_0^{\text{rx}}$, we can compute the gradients of the predicted channels $\nabla_{\boldsymbol{x}^{\text{rx}}} h_\theta(\boldsymbol{F}, \boldsymbol{x}_i^{\text{tx}}, \boldsymbol{x}_0^{\text{rx}})$. Using gradient descent, we iteratively refine the receiver position to solve $\hat{\boldsymbol{x}}^{\text{rx}} = \arg\min_{\boldsymbol{x}^{\text{rx}}} \sum_i \|h_\theta(\boldsymbol{F}, \boldsymbol{x}_i^{\text{tx}}, \boldsymbol{x}^{\text{rx}}) - h_i\|^2$.

**Probabilistic inference with diffusion models.** While a predictive model of the signal can serve as a versatile neural simulator, it has two shortcomings. First, solving an inverse problem through gradient descent requires a sizable computational cost for every problem instance. Second, predictive models are deterministic and do not allow us to model stochastic forward processes or express the inherent uncertainty in inverse problems. To overcome this, we draw inspiration from the inverse problem solving capabilities of diffusion models using guidance (Chung et al., 2022; Gloeckler et al., 2024). We hereby follow the DDPM framework and use a Wi-GATr model as score estimator (denoising network) to learn a joint distribution $p_\theta(\boldsymbol{F}, \boldsymbol{x}^{\text{tx}}, \boldsymbol{x}^{\text{rx}}, h)$ between 3D environment mesh $\boldsymbol{F}$, transmitter $\boldsymbol{x}^{\text{tx}}$, receiver $\boldsymbol{x}^{\text{rx}}$, and channel $h$, for a single transmitter-receiver pair (see Fig. 1b). See Appendix E.2 for more details.

## 4 New datasets with diverse scene geometry

To show the importance of geometry and symmetries in wireless signal propagation, we require a dataset that comprises complex signal interactions with diverse geometries. While several datasets of wireless simulations and measurements exist (Orekondy et al., 2022b; Zhang et al., 2023; Alkhateeb, 2019; Alkhateeb et al., 2023), they either do not include geometric information, lack scene diversity or the signal predictions are not realistic. Therefore, we generate two datasets that feature indoor scenes and channel information at a frequency of 3.5 GHz using *Wireless InSite*, a state-of-the-art ray-tracing simulator (Remcom).[3] We focus on indoor scenes as transmission plays a stronger role than outdoors. The datasets provide detailed characteristics for each path between Tx and Rx, such

---

[2]We also experimented with a reciprocity-equivariant variation of the architecture, but that led to a marginally worse performance without a significant gain in sample efficiency.

[3]The datasets are available at https://github.com/Qualcomm-AI-research/WiInSim.

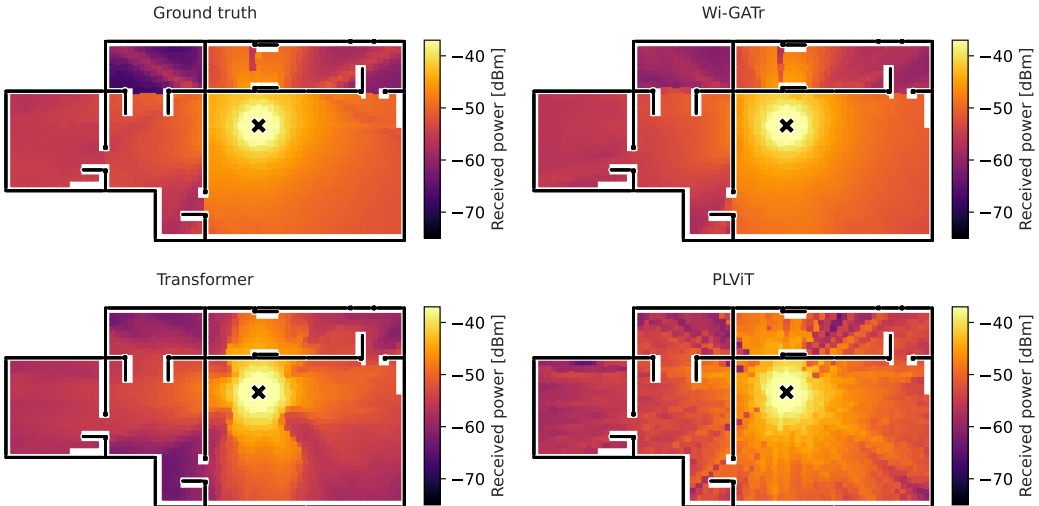

**Figure 2: Qualitative signal prediction results.** We show a single floor plan from the `WiPTR` test set. The black lines indicate the walls and doors, the colors show the received power as a function of the transmitter location (brighter colours mean a stronger signal). The transmitting antenna is shown as a black cross. The $z$ coordinates of transmitter and receiver are all fixed to the same height. We compare the ground-truth predictions (top left) to the predictions from different predictive models, each trained on only 100 `WiPTR` floor plans. Wi-GATr is able to generalize to this unseen floor plan even with such a small training set.

as gain, delay, angle of departure and arrival at Tx/Rx, and the electric field at the receiver itself, which allows users to compute various quantities of interest themselves. See Appendix D for more details.

**Wi3R dataset.** Our first dataset focuses on diversity of room layouts and their impact on the signal. We take 5000 layouts from `Wi3Rooms` (Orekondy et al., 2022b) and randomly sample 3D Tx positions and Rx positions. In Appendix D we provide more details and define training, validation, and test splits as well as an out-of-distribution set to test the robustness of different models.

**WiPTR dataset.** To increase complexity, we generate a second, more varied, realistic dataset that includes material properties, varying number of faces, and varying layout dimensions. The floor layouts are based on the `ProcTHOR-10k` dataset for embodied AI research (Deitke et al., 2022). This dataset consists of 12k different floor layouts, split into training, test, validation, and OOD sets as described in Appendix D. `WiPTR` stands out among wireless datasets in terms of geometric diversity of scenes, which enables the geometric deep learning community to study symmetries in electromagnetic signal propagation. Additionally, since it is based on `ProcTHOR-10k`, it can provide wireless signals as a novel modality for embodied AI research.

## 5 EXPERIMENTS

In our experiments, we provide empirical evidence for the benefits of the inductive bias of our equivariant architecture, as well as the inference speed, differentiability, and expressiveness of fully-learnt neural simulation surrogates on simulated and real-world data. In addition, we demonstrate probabilistic inference through a diffusion model.

### 5.1 SIGNAL STRENGTH PREDICTION ON SIMULATED DATA

On the simulated data, we focus on the prediction of the time-averaged non-coherent received power $h = \sum_p |a_p|^2$, where $|a_p|^2$ is the received power of each independent ray tracing path $p$. We train surrogates $h_\theta(\boldsymbol{F}, \boldsymbol{x}^{\text{tx}}, \boldsymbol{x}^{\text{rx}})$ that predict the power as a function of the Tx position and orientation $\boldsymbol{x}^{\text{tx}}$, Rx position and orientation $\boldsymbol{x}^{\text{rx}}$, and 3D environment mesh $\boldsymbol{F}$, on both the `Wi3R` and `WiPTR` datasets. All models are trained with reciprocity augmentation, i.e., randomly flipping Tx and Rx

| | `Wi3R` dataset | | | | `WiPTR` dataset | | |
|---|---|---|---|---|---|---|---|
| | Wi-GATr (ours) | Transf. | SEGNN | PLViT | Wi-GATr (ours) | Transf. | PLViT |
| *In distribution* | | | | | | | |
| Unseen Rx positions | **0.63** | 1.14 | 0.92 | 4.52 | **0.39** | 0.62 | 1.27 |
| Unseen floor plans | **0.74** | 1.32 | 1.02 | 4.81 | **0.41** | 0.69 | 1.28 |
| *Symmetry transformations* | | | | | | | |
| Rotation | **0.74** | 78.68 | 1.02 | 4.81 | **0.41** | 38.51 | 1.28 |
| Translation | **0.74** | 64.05 | 1.02 | 4.81 | **0.41** | 4.96 | 1.28 |
| Permutation | **0.74** | 1.32 | 1.02 | 4.81 | **0.41** | 0.69 | 1.28 |
| Reciprocity | **0.80** | 1.32 | 1.01 | 10.15 | **0.41** | 0.69 | 1.28 |
| *Out of distribution* | | | | | | | |
| OOD layout | 7.03 | 14.06 | **2.34** | 5.89 | **0.43** | 0.86 | 1.23 |

**Table 1: Signal prediction results.** We show the mean absolute error on the received power in dB (lower is better, best in bold). **Top**: In-distribution performance. **Middle**: Generalization under symmetry transformations. **Bottom**: Generalization to out-of-distribution settings. In almost all settings, Wi-GATr is the highest-fidelity surrogate model.

labels during training. This improves data efficiency, especially for the transformer baseline.

**Baselines.** In addition to our Wi-GATr model, described in Sec. 3, we train several baselines. The first is a vanilla transformer (Vaswani et al., 2017), based on the same inputs and tokenization of the wireless scene, but without the geometric inductive biases. To ablate the effect of our tokenization scheme, we also test a transformer without it, which represents the scene as a simple sequence of 3D positions. Next, we compare to the E(3)-equivariant SEGNN (Brandstetter et al., 2022b) on `Wi3R`. On `WiPTR`, training this model was impractical as it required too much memory (>80 GB). As state-of-the-art method for image-based wireless channel modelling, we train a PLViT model (Hehn et al., 2023). We have also considered WiNeRT (Orekondy et al., 2022b) and Sionna RT (Hoydis et al., 2022) as baselines, but found them unsuited for the problem setup as explained in Appendix E. More details on our experiment setup and the baselines are also given in Appendix E.

**In-distribution and out-of-distribution performance.** In Fig. 2 we illustrate the prediction task on a `WiPTR` floor plan. We show signal predictions for the simulator as well as for surrogate models trained on only 100 floor plans. While the transformer and PLViT show memorization artifacts, Wi-GATr is able to capture the propagation pattern, although this floor plan was not seen during training.

In Tbl. 1 we compare surrogate models trained on the full `Wi3R` and `WiPTR` datasets. Both when evaluating unseen Rx positions on seen training floor plans as well as when evaluating on new scenes unseen during training, Wi-GATr offers the highest-fidelity approximation of the simulator. Wi-GATr as well as the equivariant baselines are by construction robust to symmetry transformations, while the performance of a vanilla transformer degrades substantially. All methods but SEGNN struggle to generalize to an OOD setting on the `Wi3R` dataset. This is not surprising given that the training samples are so similar to each other. On the more diverse `WiPTR` dataset, Wi-GATr is almost perfectly robust under domain shift.

**Data efficiency.** Next, we study the data efficiency of the different surrogates in Fig. 3. Wi-GATr is more data-efficient than any other method with the exception of the E(3)-equivariant SEGNN, which performs similarly well for a small number of training samples. This confirms that equivariance is a useful inductive bias when data is scarce. But Wi-GATr scales better than SEGNN to larger number of samples, showing that our architecture combines the small-data advantages of strong inductive biases with the large-data advantages of a transformer architecture. We find clear benefits of our wireless tokenizer when comparing the transformer baselines with and without the tokenizer.

**Inference speed.** One of the advantages of neural surrogates is their test-time speed. Both Wi-GATr and a transformer are over a factor of 20 faster than the ground-truth ray tracer (see Appendix E).

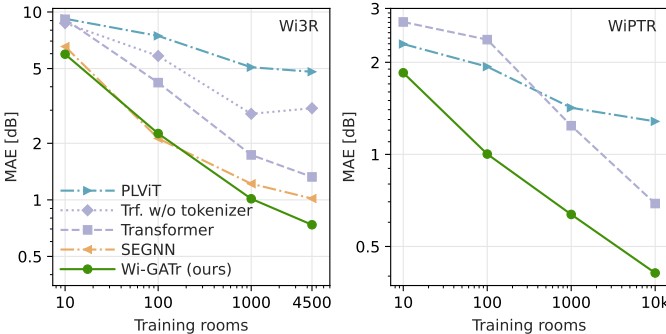 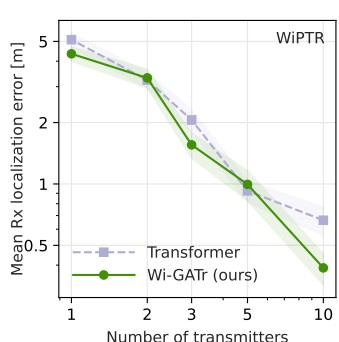

**Figure 3: Signal prediction.** We show the mean absolute error on the received power as a function of the training data on `Wi3R` (left) and `WiPTR` (right). Wi-GATr outperforms the transformer and PLViT baselines at any amount of training data, and scales better to large data or many tokens than SEGNN.

**Figure 4: Rx localization error**, as a function of the number of Tx. Lines and error band show mean and its standard error over 240 measurements.

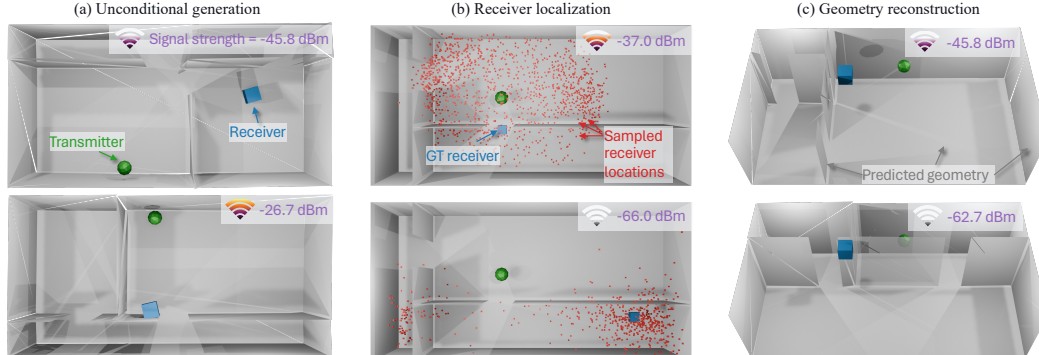

**Figure 5: Probabilistic modelling.** We formulate various tasks as sampling from the unconditional or conditional densities of a single diffusion model. **(a)**: Unconditional sampling of wireless scenes $p(\boldsymbol{F}, \boldsymbol{x}^{\mathrm{tx}}, \boldsymbol{x}^{\mathrm{rx}}, h)$. **(b)**: Receiver localization as conditional sampling from $p(\boldsymbol{x}^{\mathrm{rx}}|\boldsymbol{F}, \boldsymbol{x}^{\mathrm{tx}}, h)$ for two different values of $h$ and $\boldsymbol{x}^{\mathrm{rx}}$. **(c)**: Geometry reconstruction as conditional sampling from $p(\boldsymbol{F}_u|\boldsymbol{F}_k, \boldsymbol{x}^{\mathrm{tx}}, \boldsymbol{x}^{\mathrm{rx}}, h)$ for two different values of $h$, keeping $\boldsymbol{x}^{\mathrm{tx}}, \boldsymbol{x}^{\mathrm{rx}}, \boldsymbol{F}_k$ fixed.

## 5.2 INVERSE RECEIVER LOCALIZATION

We show how differentiable surrogates let us solve inverse problems, focusing on the problem of receiver (Rx) localization. We infer the Rx position with the predictive surrogate models by optimizing through the neural surrogate of the simulator as discussed in Sec. 3.3. The performance of our surrogate models is shown in Fig. 4 and Appendix E.[4] The two neural surrogates achieve a similar performance when only one or two transmitters are available, a setting in which the receiver position is ambiguous. With more measurements, Wi-GATr lets us localize the transmitter more precisely.

## 5.3 INVARIANT DIFFUSION FOR INVERSE PROBLEMS

In the following, we consider the problem of signal prediction as well as the inverse problems of receiver localization and geometry reconstruction. All three are instances of sampling from conditional densities: $h \sim p_\theta(h|\boldsymbol{F}, \boldsymbol{x}^{\mathrm{tx}}, \boldsymbol{x}^{\mathrm{rx}})$, $\boldsymbol{x}^{\mathrm{rx}} \sim p_\theta(\boldsymbol{x}^{\mathrm{rx}}|\boldsymbol{F}, \boldsymbol{x}^{\mathrm{tx}}, h)$ and $\boldsymbol{F} \sim p_\theta(\boldsymbol{F}|\boldsymbol{x}^{\mathrm{tx}}, \boldsymbol{x}^{\mathrm{rx}}, h)$, respectively. We obtain the conditional densities by inpainting during the sampling from the joint densities (Sec. 3.3; Sohl-Dickstein et al. (2015)). To do so, we train a diffusion model using a Wi-GATr backbone with the DDPM pipeline and 1000 denoising steps. We additionally apply conditional masking strategies to improve inpainting performance. See Appendix C for a detailed description of our diffusion model, the masking protocol, as well a discussion on equivariant generative modelling. For

---

[4] Neither the SEGNN nor PLViT baselines are fully differentiable with respect to object positions when using the official implementations from Refs. (Brandstetter et al., 2022a; Hehn et al., 2023). We were therefore not able to accurately infer the transmitter positions with these architectures.

comparison, we study a transformer baseline and a transformer trained on the same data augmented with random rotations. More details on the model architectures can be found in Appendix E.

We qualitatively show results for this approach in Figs. 1 and 5. All predictions are probabilistic, which allows our model to express uncertainty in ambiguous inference tasks. In Fig. 5 we visualize different conditional samples. For each example, we sample from the same trained Wi-GATr diffusion model, while only varying the conditioning mask. In the case of receiver localization, we sample a single batch of points $(\boldsymbol{F}, \boldsymbol{x}^{\text{tx}}, \boldsymbol{x}_i^{\text{rx}}, h)_{i=1}^{B}$, where $\boldsymbol{F}$, $\boldsymbol{x}^{\text{tx}}$ and $h$ are fixed and $B$ denotes the batch size. In this case, we observe that the model learns multimodal densities corresponding to plausible receiver locations. Furthermore, we see that the model is sensitive to a change in signal strength $h$, with large values of $h$ resulting in receiver locations close to the given transmitter $\boldsymbol{x}^{\text{tx}}$, whereas small values of $h$ result in sample locations that avoid direct line of sight with the transmitter. Similarly, when reconstructing geometry, the model will sample diverse floor plans as long as they are consistent with the transmitted signal, see the right panel of Fig. 5. Additional results on signal and geometry prediction are given in Appendix E.2.

We quantitatively evaluate these models through the variational lower bound on the log likelihood of test data under the model. To further analyze the effects of equivariance, we test the model both on canonicalized scenes, in which all walls are aligned with the $x$ and $y$ axis, and scenes that are arbitrarily rotated. The results in Tbl. 2 show that Wi-GATr outperforms the transformer baseline across all three tasks, even in the canonicalized setting or when the transformer is trained with data augmentation. The gains of Wi-GATr are particularly clear on the signal prediction and receiver localization problems.

| | Wi-GATr (ours) | Transformer default | data augm. |
|---|---|---|---|
| *Canonicalized scenes* | | | |
| Signal pred. | **1.62** | 3.00 | 15.66 |
| Receiver loc. | **3.64** | 8.28 | 14.42 |
| Geometry reco. | **-3.95** | -3.61 | -2.10 |
| *Scenes in arbitrary rotations* | | | |
| Signal pred. | **1.62** | 9.57 | 17.65 |
| Receiver loc. | **3.64** | 105.68 | 14.45 |
| Geometry reco. | **-3.95** | 389.34 | -2.34 |

Table 2: **Probabilistic modelling results**. We show variational upper bounds on the negative log likelihood for different conditional inference tasks (lower is better, best in bold).

## 5.4 Comparison to differentiable ray tracing on real-world data

The utility of wireless channel simulation is ideally evaluated on real data. Unfortunately, the scarcity of available measurement data limits evaluation to simple scenarios and prevents thorough testing of the geometric generalization capabilities. Yet, we aim to highlight limitations of *hybrid* ray tracing methods compared to our *fully-learnt* approach. To this end, we replicate the evaluation setup of Hoydis et al. (2023) on the DICHASUS dataset (Euchner et al., 2021).

The dataset consists of a single hallway with receiver arrays at each end and an occluded corridor in between. During collection, a robot-mounted transmitter was moving through the hallway, covering most of the area, with accurate location data coming from a tachymeter. We train our model to predict the average received power and the average delay spread for each receiver array, using the same data splits and evaluation scheme as in Hoydis et al. (2023). We report the mean absolute (logarithmic) error

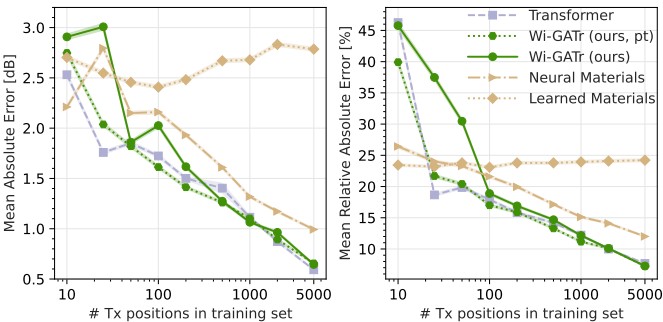

Figure 6: **Signal prediction on real measurements.** As a function of the training data size, we show the mean absolute (logarithmic) error on the received power (left) and the relative absolute error on the delay spread (right).

on the average received power and the relative absolute error on the average delay spread (see equations 44 & 45 in Hoydis et al. (2023)). For this purpose, we can consider each array as a single receiver without modifications to our Wi-GATr backbone. As baselines, we run their *hybrid* ap-

proaches named *Learned Materials* and *Neural Materials*.

In Figure 6, we show the performance in both prediction tasks with respect to the number of transmitter locations used during training. Each transmitter location corresponds to two data samples, one per receiver. The Learned Materials approach maintains consistent performance due to model bias, making it robust with limited data but less flexible with larger datasets. Neural Materials also benefits from model bias with little data but, thanks to its flexible material model, can leverage data to overcome other model and geometry limitations. Our *fully-learnt* approaches Wi-GATr and the Transformer outperform both *hybrid* models as the dataset size grows, while their drawback of higher variance in the small data regime can be stabilized through pretraining on `WiPTR` (indicated by "pt"). We find that using a Transformer with the Wi-GATr tokenization scheme slightly outperforms GATr in the small data regime. However, we attribute this to simplicity of the dataset, where symmetry transformations of 3D space are neither observed nor required.

## 6 CONCLUSION

We developed a class of neural surrogates grounded in geometric representations and strong inductive biases. They are based on the Wi-GATr backbone, consisting of a new tokenization scheme for wireless scenes together with an $E(3)$-equivariant transformer architecture. In our experiments, we demonstrated the benefits of the inductive bias, inference speed, differentiability, and expressiveness of our fully-learnt equivariant neural simulation surrogate on simulated and real-world data. In addition, we showed how our backbone can be used in a diffusion model for probabilistic inference.

**Limitations.** Our paper presents a step towards fully-learnt wireless simulations, which has natural trade-offs when compared to model-based and hybrid ray tracers. Unlike model-based approaches (which require no training data), learnable approaches depend on reasonably-sized amounts of training data and hence involve data collection overhead. Furthermore, model-based approaches are highly configurable (e. g., scene parameterization, antenna patterns, controlling precision), whereas our approach is less configurable with many configuration choices learnt implicitly using training data. Finally, we emphasize that we are not proposing a drop-in replacement of a model-based ray tracer, but rather take a step towards the overarching goal of learning wireless simulations from data.

**Acknowledgements.**

We thank Suresh Sharma, Pim de Haan, Maximilian Arnold, and Jens Petersen for fruitful discussions.

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

## A  GEOMETRIC ALGEBRA

As representation, Wi-GATr uses the projective geometric algebra $\mathbb{G}_{3,0,1}$. Here we summarize key aspects of this algebra and define the canonical embedding of geometric primitives in it. For a precise definition and pedagogical introduction, we refer the reader to Dorst (2020).

**Geometric algebra.**  A geometric algebra $\mathbb{G}_{p,q,r}$ consists of a vector space together with a bilinear operation, the *geometric product*, that maps two elements of the vector space to another element of the vector space.

The elements of the vector space are known as *multivectors*. Their space is constructed by extending a base vector space $\mathbb{R}^d$ to lower orders (scalars) and higher-orders (bi-vectors, tri-vectors, ...). The algebra combines all of these orders (or *grades*) in one $2^d$-dimensional vector space. From a basis for the base space, for instance $(e_1, e_2, e_3)$, one can construct a basis for the multivector space. A multivector expressed in that basis then reads, for instance for $d = 3$, $x = x_\emptyset + x_1 e_1 + x_2 e_2 + x_3 e_3 + x_{12} e_1 e_2 + x_{13} e_1 e_3 + x_{23} e_2 e_3 + x_{123} e_1 e_2 e_3$.

The geometric product is fully defined by bilinearity, associativity, and the condition that the geometric product of a vector with itself is equal to its norm. The geometric product generally maps between different grades. For instance, the geometric product of two vectors will consist of a scalar, the inner product between the vectors, and a bivector, which is related to the cross-product of $\mathbb{R}^3$. In particular, the conventional basis elements of grade $k > 1$ are constructed as the geometric product of the vector basis elements $e_i$. For instance, $e_{12} = e_1 e_2$ is a basis bivector. From the defining properties of the geometric products it follows that the geometric product between orthogonal basis elements is antisymmetric, $e_i e_j = -e_j e_i$. Thus, for a $d$-dimensional basis space, there are $\binom{d}{k}$ independent basis elements at grade $k$.

**Projective geometric algebra.**  To represent three-dimensional objects including absolute positions, we use a geometric algebra based on a base space with $d = 4$, adding a *homogeneous coordinate* to the 3D space.[5] We use a basis $(e_0, e_1, e_2, e_3)$ with a metric such that $e_0^2 = 0$ and $e_i^2 = 1$ for $i = 1, 2, 3$. The multivector space is thus $2^4 = 16$-dimensional. This algebra is known as the projective geometric algebra $\mathbb{G}_{3,0,1}$.

**Canonical embedding of geometric primitives.**  In $\mathbb{G}_{3,0,1}$, we can represent geometric primitives as follows:

- Scalars (data that do not transform under translation, rotations, and reflections) are represented as the scalars of the multivectors (grade $k = 0$).
- Oriented planes are represented as vectors ($k = 1$), encoding the plane normal as well as the distance from the origin.
- Lines or directions are represented as bivectors ($k = 2$), encoding the direction as well as the shift from the origin.
- Points or positions are represented as trivectors ($k = 3$).

For more details, we refer the reader to Tbl. 1 in Brehmer et al. (2023), or to Dorst (2020).

## B  WIRELESS GEOMETRIC ALGEBRA TOKENIZER

Table 3 outlines how the individual scene parameters are embedded into Geometric Algebra.

## C  DIFFUSION MODEL

Formally, we employ the standard DDPM framework Song et al. (2021) to train a latent variable model $p_\theta(\mathbf{x}_0) = \int p_\theta(\mathbf{x}_{0:T}) d_{\mathbf{x}_{1:T}}$, where $\mathbf{x}_0 = [\boldsymbol{F}, \boldsymbol{x}^{\text{tx}}, \boldsymbol{x}^{\text{rx}}, h]$ denotes the joint vector of variables following the dataset distribution $p_{data}(\mathbf{x}_0)$. In DDPM, the latent variables $\mathbf{x}_{1:T}$ are noisy versions of the original data, defined by a discrete forward noise process $q(\mathbf{x}_t|\mathbf{x}_{t-1}) =$

---

[5]A three-dimensional base space is not sufficient to represent absolute positions and translations acting on them in a convenient form. See Dorst (2020); Ruhe et al. (2023); Brehmer et al. (2023) for an in-depth discussion.

$\mathcal{N}\left(\mathbf{x}_t; \sqrt{1 - \beta_t}\mathbf{x}_{t-1}, \beta_t\mathbf{I}\right)$ and $\beta_i > 0$. We approximate the reverse distribution $q(\mathbf{x}_{t-1}|\mathbf{x}_t)$ with $p_\theta(\mathbf{x}_{t-1}|\mathbf{x}_t) = \mathcal{N}(\mathbf{x}_{t-1}; \boldsymbol{\mu}_\theta(\mathbf{x}_t, t), \sigma_t^2\mathbf{I})$. The variance $\sigma^2$ is typically chosen to match the variance of $q(\mathbf{x}_{t-1}|\mathbf{x}_t, \mathbf{x}_0)$, which has been shown to follow a closed-form normal distribution Ho et al. (2020). Furthermore, instead of parametrizing $\boldsymbol{\mu}_\theta$ to predict the mean $\tilde{\boldsymbol{\mu}}$ of $q(\mathbf{x}_{t-1}|\mathbf{x}_t, \mathbf{x}_0)$, we opt to predict the denoised sample $\hat{\mathbf{x}}_0 = f_\theta(\mathbf{x}_t, t)$. For a discussion of these different parametrizations we refer to (Ho et al., 2020, Section 3.2). The forward and backward distributions $q$ and $p$ form a variational auto-encoder Kingma & Welling (2014) which can be trained with a variational lower bound loss. Using the above parametrization of $p_\theta(\mathbf{x}_{t-1}|\mathbf{x}_t)$, however, allows for a simple approximation of this lower bound by training on an MSE objective $\mathcal{L} = \mathbb{E}_{\mathbf{x}_t, \mathbf{x}_0}\left[||f_\theta(\mathbf{x}_t, t) - \mathbf{x}_0||^2\right]$ which resembles denoising score matching Vincent (2011). We provide an outline of the training algorithm in Algorithm 1, which closely follows the proposed scheme of Ho et al. (2020).

To predict denoised samples using $f_\theta(\mathbf{x}_t, t)$, we pass the raw representation of $\mathbf{x}_t$ through the wireless GA tokenizer of Wi-GATr and, additionally, we embed the scalar $t$ through a learned timestep embedding Peebles & Xie (2022). The embedded timesteps can then be concatenated along the scalar channels in the GA representation in a straightforward manner. Similar to GATr Brehmer et al. (2023), the neural network outputs a prediction in the GA representation, which is subsequently converted to the original latent space. Note that this possibly simplifies the learning problem, as the GA representation is inherently higher dimensional than our diffusion space with the same dimensionality as $\mathbf{x}_0$.

**Related Work.** Diffusion models (Sohl-Dickstein et al., 2015; Ho et al., 2020; Song et al., 2021) are a class of generative models that iteratively invert a noising process. They have become the de-facto standard in image and video generation (Ramesh et al., 2022; Ho et al., 2022). Recently, they have also shown to yield promising results in the generation of spatial and sequential data, such as in planning (Janner et al., 2022) and puzzle solving (Hossieni et al., 2024). Aside from their generative modelling capabilities, diffusion models provide a flexible way for solving inverse problems (Chung et al., 2022; Lugmayr et al., 2022; Gloeckler et al., 2024) through multiplication with an appropriate likelihood term (Sohl-Dickstein et al., 2015). Furthermore, by combining an invariant prior distribution with an equivariant denoising network, one obtains equivariant diffusion models (Köhler et al., 2020). These yield a sampling distribution that assigns equal probability to all symmetry transformations of an object, which can improve performance and data efficiency in symmetry problems like molecule generation (Hoogeboom et al., 2022) and planning (Brehmer et al., 2024). We demonstrate similar benefits in modelling wireless signal propagation.

**Equivariant generative modelling.** A diffusion model with an invariant base density and an equivariant denoising network defines an invariant density, but equivariant generative modelling has some subtleties Köhler et al. (2020). Because the group of translations is not compact, we cannot define a translation-invariant base density. Previous works have circumvented this issue by performing diffusion in the zero center of gravity subspace of euclidean space Hoogeboom et al.

| Data type | Input parameterization | Tokenization | Channels ($\mathbb{G}_{3,0,1}$ embedding) |
|---|---|---|---|
| 3D environment $\boldsymbol{F}$ | • Triangular mesh | 1 token per mesh face | • Mesh face center (point) |
| | | | • Vertices (points) |
| | | | • Mesh face plane (oriented plane) |
| | • Material classes | | • One-hot material emb. (scalars) |
| Antenna $\boldsymbol{x}^{\mathrm{tx}}$ / $\boldsymbol{x}^{\mathrm{rx}}$ | • Position | 1 token per antenna | • Position (point) |
| | • Orientation | | • Orientation (direction) |
| | • Receiving / transmitting | | • One-hot type embedding (scalars) |
| | • Additional characteristics | | • Characteristics (scalars) |
| Channel $h$ | • Antennas | 1 token per link | • Tx position (point) |
| | | | • Rx position (point) |
| | | | • Tx-Rx vector (direction) |
| | • Received power | | • Normalized power (scalar) |
| | • Phase, delay, . . . | | • Additional data (scalars) |

**Table 3: Wireless GA tokenizer.** We describe how the mesh parameterizing the 3D environment and the information about antennas and their links are represented as a sequence of geometric algebra tokens. The mathematical representation of $\mathbb{G}_{3,0,1}$ primitives like points or orientated planes is described in Appendix A.

---

**Algorithm 1** Diffusion Wi-GATr Training

1: **Input** $\phi$, **Initialize** $\theta$                          ▷ Specify mask probability distribution
2: **repeat**
3:      $\mathbf{x}_0 \sim p_{data}(\mathbf{x}_0)$
4:      $t \sim \mathrm{Uniform}(\{1, \ldots, T\})$
5:      $\mathbf{x}_t \sim q(\mathbf{x}_t | \mathbf{x}_0)$                             ▷ Create noisy sample
6:      $\mathbf{m} \sim p_{mask}(\mathbf{m}; \boldsymbol{\phi})$                    ▷ Sample binary conditioning mask
7:      $\mathbf{x}_t^{\mathbf{m}} = \mathbf{m} \odot \mathbf{x}_t + (1 - \mathbf{m}) \odot \mathbf{x}_0$             ▷ Apply conditioning per token
8:      Take gradient descent step on
         $\nabla_\theta \left[ || \mathbf{m} \odot \left( f_\theta(\mathbf{x}_t^{\mathbf{m}}, t, \mathbf{m}) - \mathbf{x}_0 \right) ||^2 \right]$
9: **until** converged

---

**Algorithm 2** Diffusion Wi-GATr Sampling

1: **Input** $\mathbf{m}$, $\mathbf{x}_0^{\mathbf{m}}$                        ▷ Specify conditioning mask and conditions
2: $\mathbf{x}_T \sim \mathcal{N}(0, \mathbf{I})$
3: **for** $t = T \ldots 0$ **do**
4:      $\mathbf{x}_t = \mathbf{m} \odot \mathbf{x}_t + (1 - \mathbf{m}) \odot \mathbf{x}_0^{\mathbf{m}}$              ▷ Apply conditioning per token
5:      $\hat{\mathbf{x}}_0 = f_\theta(\mathbf{x}_t, t, \mathbf{m})$                     ▷ Predict denoised sample
6:      $\mathbf{x}_{t-1} \sim q\left(\mathbf{x}_{t-1} | \mathbf{x}_t, \hat{\mathbf{x}}_0\right)$       ▷ Compute predicted sample at $t-1$ from $\hat{\mathbf{x}}_0$
7: **end for**
8: **return** $x_0$

---

(2022). However, we found that directly providing the origin as an additional input to the denoising network also resulted in good performance, at the cost of full E(3) equivariance. We also choose to generate samples in the convention where the $z$-axis represents the direction of gravity and positive $z$ is "up"; we therefore provide this direction of gravity as an additional input to our network.

**Unifying forward prediction and inverse problems as conditional sampling.** A diffusion model trained to learn the joint density $p_\theta(\boldsymbol{F}, \boldsymbol{x}^{\mathrm{tx}}, \boldsymbol{x}^{\mathrm{rx}}, h)$ does not only allow us to generate unconditional samples of wireless scenes, but also lets us sample from various conditionals: given a partial wireless scene, we can fill in the remaining details, in analogy to how diffusion models for images allow for inpainting. To achieve this, we use the conditional sampling algorithm proposed by Sohl-Dickstein et al. (2015): at each step of the sampling loop, we fix the conditioning variables to their known values before feeding them into the denoising network. This algorithm lets us solve signal prediction (sampling from $p_\theta(h|\boldsymbol{F}, \boldsymbol{x}^{\mathrm{tx}}, \boldsymbol{x}^{\mathrm{rx}})$), receiver localization (from $p_\theta(\boldsymbol{x}^{\mathrm{rx}}|\boldsymbol{F}, \boldsymbol{x}^{\mathrm{tx}}, h)$), geometry reconstruction (from $p_\theta(\boldsymbol{F}_u|\boldsymbol{F}_k, \boldsymbol{x}^{\mathrm{tx}}, \boldsymbol{x}^{\mathrm{rx}}, h)$), or any other inference task in wireless scenes. We thus unify "forward" and "inverse" modelling in a single algorithm. Each approach is probabilistic, enabling us to model uncertainties. This is important for inverse problems, where measurements often underspecify the solutions.

**Masking strategies.** In principle, the unconditional diffusion objective should suffice to enable test-time conditional sampling. In practice, we find that we can improve the conditional sampling performance with two modifications. First, we combine training on the unconditional diffusion objective with conditional diffusion objectives. For the latter, we randomly select tokens to condition on and evaluate the diffusion loss only on the remaining tokens. Second, we provide the binary conditioning mask as an additional input to the denoising model. Non-zero entries in the mask indicate what data we wish to generate, while zero entries are the ones we condition on. We sample masks from a discrete categorical distribution with probabilities $\boldsymbol{\phi} = (0.2, 0.3, 0.2, 0.3)$ corresponding to masks for unconditional, signal, receiver and mesh prediction, respectively. If we denote this distribution over masks as $p_{mask}(\mathbf{m}; \boldsymbol{\phi})$, the modified loss function then reads as $\mathcal{L} = \mathbb{E}_{\mathbf{m} \sim p_{mask}(\mathbf{m};\boldsymbol{\phi}), \mathbf{x}_t, \mathbf{x}_0} \left[ || \mathbf{m} \odot f_\theta(\mathbf{x}_t^{\mathbf{m}}, t, \mathbf{m}) - \mathbf{m} \odot \mathbf{x}_0 ||^2 \right]$, where $\mathbf{x}_t^{\mathbf{m}}$ is equal to $\mathbf{x}_0$ along the masked tokens according to $\mathbf{m}$. To see how the masks are used during training and inference, we refer to Algorithm 1 and 2. We note that while any arbitrary mask $\mathbf{m}$ of adequate dimensionality can be passed into the model, we only test on masks $\mathbf{m} \in \mathrm{supp}(p_{mask}(\mathbf{m}; \phi))$. That is, we only sample from the unconditional distribution or a conditional distribution corresponding to signal prediction, receiver localization and geometry reconstruction, while respectively keeping all other tokens fixed.

# D  DATASETS

Table 4 summarizes major characteristics of the two datasets. In the following we explain more details on data splits and generation.

**Wi3R dataset.** Based on the layouts of the Wi3Rooms dataset by Orekondy et al. (2022b), we run simulations for 5000 floor layouts that are split into training (4500), validation (250), and test (250). These validation and test splits thus represent generalization across unseen layouts, transmitter, and receiver locations. From the training set, we keep 10 Rx locations as additional test set to evaluate generalization only across unseen Rx locations. To evaluate the generalization performance, we also introduce an out-of-distribution (OOD) set that features four rooms in each of the 250 floor layouts. In all layouts, the interior walls are made of brick while exterior walls are made of concrete. The Tx and Rx locations are sampled uniformly within the bounds of the floor layouts (10m $\times$ 5m $\times$ 3m).

**WiPTR dataset.** Based on the floor layouts in the ProcTHOR-10k dataset for embodied AI research Deitke et al. (2022), we extract the 3D mesh information including walls, windows, doors, and door frames. The layouts comprise between 1 to 10 rooms and can cover up to 600 m$^2$. We assign 6 different dielectric materials for different groups of objects (see Tbl. 5). The 3D Tx and Rx locations are randomly sampled within the bounds of the layout. The training data comprises 10k floor layouts, while test and validation sets each contain 1k unseen layouts, Tx, and Rx locations. Again, we introduce an OOD validation set with 5 layouts where we manually remove parts of the walls such that two rooms become connected. While the multi-modality in combination with the ProcTHOR dataset enables further research for joint sensing and communication in wireless, our dataset set is also, to the best of our knowledge, the first large-scale 3D wireless indoor datasets suitable for embodied AI research.

# E  EXPERIMENTS

## E.1  PREDICTIVE MODELLING

**Models.** We use an Wi-GATr model that is 32 blocks deep and 16 multivector channels in addition to 32 additional scalar channels wide. We use 8 attention heads and multi-query attention. Overall, the model has $1.6 \cdot 10^7$ parameters. These settings were selected by comparing five differently sized networks on an earlier version of the Wi3R dataset, though somewhat smaller and bigger networks achieved a similar performance.

Our Transformer model has the same width (translating to 288 channels) and depth as the Wi-GATr model, totalling $16.7 \cdot 10^6$ parameters. These hyperparameters were independently selected by comparing five differently sized networks on an earlier version of the Wi3R dataset.

For SEGNN, we use representations of up to $\ell_{max} = 3$, 8 layers, and 128 hidden features. The model has $2.6 \cdot 10^5$ parameters. We selected these parameters in a scan over all three parameters, within

|                         | Wi3R      | WiPTR     |
|-------------------------|-----------|-----------|
| Total Channels          | 5M        | >5.5M     |
| Materials               | 2         | 6         |
| Transmitters per layout | 5         | 1-15      |
| Receivers per layout    | 200       | Up to 200 |
| Floor layouts           | 5k        | 12k       |
| Simulated frequency     | 3.5 GHz   | 3.5 GHz   |
| Reflections             | 3         | 6         |
| Transmissions           | 1         | 3         |
| Diffractions            | 1         | 1         |
| Strongest paths retained| 25        | 25        |
| Antennas                | Isotropic | Isotropic |
| Waveform                | Sinusoid  | Sinusoid  |

**Table 4:** Dataset details and simulation settings for dataset generation.

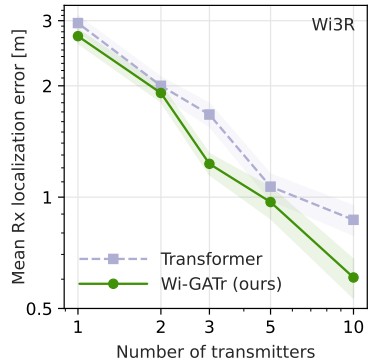

**Figure 7: Rx localization error**, as a function of the number of Tx. Lines and error band show mean and its standard error over 240 measurements.

the ranges used in Brandstetter et al. (2022b).

The PLViT model is based on the approach introduced by Hehn et al. (2023). We employ the same centering and rotation strategy as in the original approach around the Tx. Further, we extend the original approach to 3 dimensions by providing the difference in $z$-direction concatenated with the 2D $x$-$y$-distance as one token. Since training from scratch resulted in poor performance, we finetuned a ViT-B-16 model pretrained on ImageNet and keeping only the red channel. This resulted in a model with $85.4 \cdot 10^7$ parameters and also required us to use a fixed image size for each dataset that ensures the entire floor layout is visible in the image data.

Finally, we attempt to compare Wi-GATr also to WiNeRT (Orekondy et al., 2022b), a neural ray tracer. However, this architecture, which was developed to be trained on several measurements on the same floor plan, was not able to achieve useful predictions on our diverse datasets with their focus on generalization across floor plans.

**Optimization.** All models are trained on the mean squared error between the model output and the total received power in dBm. We use a batch size of 64 (unless for SEGNN, where we use a smaller batch size due to memory limitations), the Adam optimizer, an initial learning rate of $10^{-3}$, and a cosine annealing scheduler. Models are trained for $5 \cdot 10^5$ steps on the `Wi3R` dataset and for $2 \cdot 10^5$ steps on the `WiPTR` dataset.

**Inference speed.** To quantify the trade-off between inference speed and accuracy of signal prediction, we compare the ray tracing simulation with our machine learning approaches. For this purpose, we evaluate the methods on a single room of the validation set with 2 different Tx locations and two equidistant grids at $z \in \{2.3, 0.3\}$ with each 1637 Rx locations.

The left panel of Fig. 8 summarizes the average inference times per link with the corresponding standard deviation. While Wireless InSite (6/3/1, i.e., 6 reflections/3 transmissions/1 diffraction) represents our method that was used to generate the ground truth data, it is also by far the slowest approach. Note that we only measure the inference speed of Wireless InSite for each Tx individually

| Object | Material name | Rel. Permittivity [1] | Conductivity [S/m] | Thickness [cm] |
|---|---|---|---|---|
| Ceiling | ITU Ceiling Board | 1.5 | 0.002148 | 0.95 |
| Floor | ITU Floor Board | 3.66 | 0.02392 | 3.0 |
| Exterior walls | Concrete | 7.00 | 0.0150 | 30.0 |
| Interior walls | ITU Layered Drywall (3 layers) | 2.94 | 0.028148 | 1.30 |
| | | 1 | 0 | 8.90 |
| | | 2.94 | 0.028148 | 1.30 |
| Doors and door frames | ITU Wood | 1.99 | 0.017998 | 3.0 |
| Windows | ITU Glass | 6.27 | 0.019154 | 0.3 |

**Table 5:** Dielectric material properties of objects in `WiPTR`.

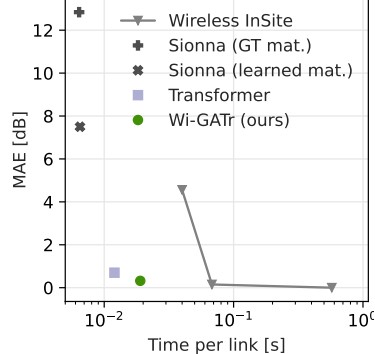 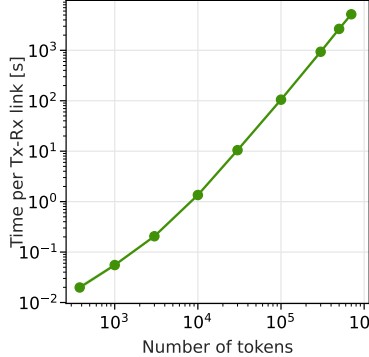

**Figure 8: Left**: **Inference wall time** vs signal prediction error per Tx/Rx prediction on the first room of the `WiPTR` validation set. **Right**: **Scaling** of the inference wall time as a function of the number of tokens (mesh faces, antennas, and links) in the scene for Wi-GATr.

without the preprocessing of the geometry. By reducing the complexity, e.g., reducing the number of allowed reflections or transmissions, of the ray tracing simulation the inference time can be reduced significantly. For example, the configuration $3/2/1$ shows a significant increase in inference speed, but at the same time we can already see that the simulation results do not match the ground truth anymore. This effect is even more pronounced for the case of Wireless InSite $3/1/1$. As of version 0.19, Sionna RT does not support any transmission and only supports first order diffraction, i.e., the path cannot include other reflection events. Therefore, in indoor scenarios such as `WiPTR`, where signals propagate through walls, Sionna is inaccurate but very fast due to the simplified model and optimized implementation. In addition, it can improve its accuracy by learning more suitable material parameters to compensate for the model simplicity, yet a fundamental gap remains. This shows that if the model is incomplete, the ground truth parameters (see Tab. 5) are not necessarily the optimal parameters. Our neural surrogates provide a favorable trade-off in terms of inference speed and prediction accuracy.

For larger scenes, the inference time of Wi-GATr grows quadratically in the number of mesh faces, just like for a standard transformer. We show this behaviour in the right panel of Fig. 8. Recently, Suk et al. (2024) have successfully demonstrated possibilities to scale GATr to large mesh sizes.

In addition, the differentiability of ML approches enables them to solve inverse problems and such as finetuning to real-world measurement data. Finetuning, often referred to as calibration, remains challenging for simulation software and will likely lead to increased MAE as the ground truth is not given by Wireless InSite itself anymore.

### E.2 PROBABILISTIC MODELLING USING DIFFUSION

**Experiment setup.** For all conditional samples involving $p(\boldsymbol{F}_u|\boldsymbol{F}_k, \boldsymbol{x}^{\text{tx}}, \boldsymbol{x}^{\text{rx}}, h)$, we always choose to set $\boldsymbol{F}_k$ to be the floor and ceiling mesh faces only and $\boldsymbol{F}_u$ to be the remaining geometry. This amounts to completely predicting the exterior walls, as well as the separating walls/doors of the three rooms, whereas the conditioning on $\boldsymbol{F}_k$ acts only as a mean to break equivariance. Since $\boldsymbol{F}$ is always canonicalized in the non-augmented training dataset, this allows for direct comparison of variational lower bounds in Tbl. 2 with the non-equivariant transformer baseline.

**Models.** For both Wi-GATr and the transformer baseline, we follow similar architecture choices as for the predictive models, using an equal amount of attention layers. To make the models timestep-dependent, we additionally employ a standard learnable timestep embedding commonly used in diffusion transformers Peebles & Xie (2022) and concatenate it to the scalar channel dimension.

**Optimization.** We use the Adam optimizer with a learning rate of $10^{-3}$ for the Wi-GATr models. The transformer models required a smaller learning rate for training stability, and thus we chose $3 \cdot 10^{-4}$. In both cases, we linearly anneal the learning rate and train for $7 \cdot 10^5$ steps with a batchsize of 64 and gradient norm clipping set to 100.

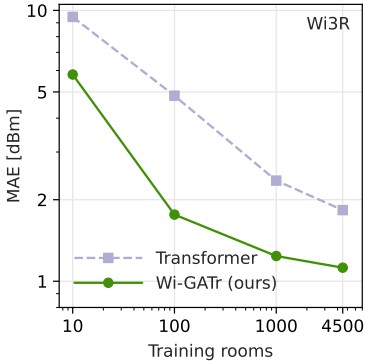

**Figure 9:** Mean absolute errors of received power as a function of number of training rooms for conditional diffusion model samples.

**Evaluation.** We use the DDIM sampler using 100 timesteps for visualizations in Fig. 5 and for the error analysis in Fig. 9. To evaluate the variational lower bound in Tbl. 2, we follow Nichol & Dhariwal (2021) and evaluate $L_{vlb} := L_0 + L_1 + \ldots L_T$, where $L_0 := -\log p_\theta(\mathbf{x}_0|\mathbf{x}_1)$, $L_{t-1} := D_{KL}(q(\mathbf{x}_{t-1}|\mathbf{x}_t, \mathbf{x}_0)||p_\theta(\mathbf{x}_{t-1}|\mathbf{x}_t))$ and $L_T := D_{KL}(q(\mathbf{x}_T|\mathbf{x}_0), p(\mathbf{x}_t))$. To be precise, for each sample $\mathbf{x}_0$ on the test set, we get a single sample $\mathbf{x}_t$ from $q$ and evaluate $L_{vlb}$ accordingly. Table 2 reports the mean of all $L_{vlb}$ evaluations over the test set.

**Additional results.** Fig. 9, shows the quality of samples from $p_\theta(h|\boldsymbol{F}, \boldsymbol{x}^{\text{tx}}, \boldsymbol{x}^{\text{rx}})$ as a function of the amount of available training data, where we average over 3 samples for each conditioning input. It is worth noting that diffusion samples have a slightly higher error than the predictive models. This shows that the joint probabilistic modelling of the whole scene is a more challenging learning task than a deterministic forward model.

To further evaluate the quality of generated rooms, we analyze how often the model generates walls between the receiver and transmitter, compared to the ground truth. Precisely, we plot the distribution of received power versus the distance of transmitter and receiver in Fig. 10 and color each point according to a line of sight test. We can see that, overall, Wi-GATr has an intersection error of $0.26$, meaning that in $26\%$ of the generated geometries, line of sight was occluded, while the true geometry did not block line of sight between receiver and transmitter. This confirms that the diffusion model correctly correlates the received power and receiver/transmitter positions with physically plausible geometries. While an error of $26\%$ is non-negligible, we note that this task involves generating the whole geometry given only a single measurement of received power, making the problem heavily underspecified. Techniques such as compositional sampling (Du et al., 2023) could overcome this limitation by allowing to condition on multiple receiver and received power measurements.

## F  DISCUSSION

Progress in wireless channel modelling is likely to lead to societal impact. Not all of it is positive. The ability to reconstruct details about the propagation environment may have privacy implications. Wireless networks are ubiquitous and could quite literally allow to see through walls. At the same time, we believe that progress in the development of wireless channel models may help to reduce radiation exposure and power consumption of wireless communication systems, and generally contribute to better and more accessible means of communication.

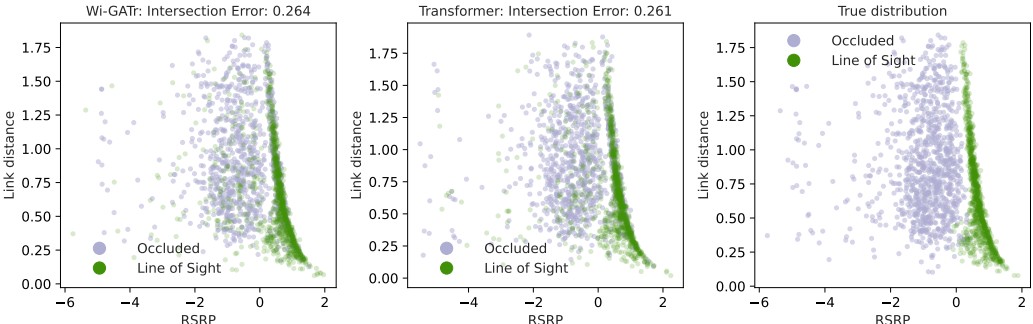

**Figure 10:** A scatter plot of normalized received power versus normalized distance between receiver and transmitter. Each point is colored depending on having line of sight between the receiver and transmitter given the room geometry. Left: The geometry used for calculating line of sight is given by conditional diffusion samples using Wi-GATr. Middle: The geometry used for calculating line of sight is given by transformer samples. Right: The geometry used for calculating line of sight is taken from the test data distribution.

