# OpenReview forum: "Differentiable and Learnable Wireless Simulation with Geometric Transformers"
_ICLR.cc/2025/Conference — ICLR 2025 Poster_

### Official Review · Reviewer_HmDB · 2024-10-20

**Soundness:** 4
**Presentation:** 3
**Contribution:** 3
**Rating:** 8
**Confidence:** 4

**Summary:**

This paper introduces Wi-GATr, a novel learnable neural surrogate for wireless channel simulation that leverages geometric primitives such as 3D surfaces, antenna positions, and orientations. The primary focus is on addressing limitations in wireless signal propagation modeling by integrating geometric algebra transformers (GATr), which enhance efficiency and accuracy. Wi-GATr is shown to outperform existing models in tasks like signal strength prediction and receiver localization, achieving significant error reductions compared to existing methods.

**Strengths:**

The paper presents a new approach, Wi-GATr, which is a neural surrogate for wireless channel modeling using Geometric Algebra Transformers, a technique not widely applied in this field. This originality sets it apart from traditional methods by addressing key limitations in differentiability and scalability. The research is supported by thorough empirical evaluations across both simulated and real-world datasets, showing substantial improvements.
The introduction of two new datasets, Wi3R and WiPTR, further enhances the credibility and reproducibility of the results. The methodology and results are clearly presented. In terms of significance, this work makes contributions to both wireless communication and machine learning.

**Weaknesses:**

1. Adam is commonly used in deep learning applications, particularly image-processing tasks. However, wireless signal modeling involves different characteristics and challenges than image data. The authors would benefit from a more detailed discussion on why Adam was chosen, especially considering the fundamental differences between wireless signal modeling and typical image tasks.

2. Wi-GATr’s generalization capabilities come from the E(3)-equivariant design of the Geometric Algebra Transformer (GATr). It would be valuable for the authors to provide more justification or discussion regarding their contribution and novelty in implementing or improving such a design.

3. While the authors introduced their own datasets (Wi3R and WiPTR), the authors should provide a clearer justification for their choice of benchmarks, and using more widely recognized simulators such as WinProp or Wireless InSite could strengthen their work.

**Questions:**

Listed above.

---

> ### Author Response · Authors · 2024-11-18
> **Response to initial reviewer comments**
>
> Thank you for your review and comments! In the following we discuss them in more detail and refer to relevant revised sections of our paper.
>
> > Why did we choose Adam?
>
> Adam is indeed a robust and efficient optimizer for training deep learning models. As Wi-GATr is a transformer, thus a deep learning model as well, the benefits of Adam during training apply here as well regardless of the exact domain. Other optimizers might also apply and lead to good Wi-GATr performance.
>
>
> > Wi-GATr’s generalization capabilities come from the E(3)-equivariant design of the Geometric Algebra Transformer (GATr). It would be valuable for the authors to provide more justification or discussion regarding their contribution and novelty in implementing or improving such a design.
>
> Thank you for this suggestion. We agree that GATr is a crucial component to this work.
> However, the architecture alone is not enough to solve practical wireless problems.
> We add both a novel tokenization scheme that allows us to represent 3D wireless problems in geometric algebra representations as well as inference algorithms, including a new approach to wireless inverse problems based on diffusion models.
> For practitioners, our release of two new datasets could be equally important.
> Beyond the technical contributions, we view our treatment of wireless channel modelling as a geometric deep learning problem as our novel key insight.
>
>
> > While the authors introduced their own datasets (Wi3R and WiPTR), the authors should provide a clearer justification for their choice of benchmarks, and using more widely recognized simulators such as WinProp or Wireless InSite could strengthen their work.
>
> We did use Wireless InSite to obtain simulation measurements in the datasets we propose.
> We have made that reference more explicit in the manuscript.
> For clear justification and detailed simulation parameters, we would like to refer to Appendix D.
> Please let us know if you are missing specific details or explanations.

---

> > ### Comment · Reviewer_HmDB · 2024-11-26
> >
> > The author answered my question to some extent. However, before some experiment results are provided, I am not very convinced by the explanation for the Adam optimizer. Besides, the fundamental differences between wireless signals and images or texts remain unexplained. I will keep the current rating now.

---

> > > ### Author Response · Authors · 2024-11-28
> > >
> > > > Besides, the fundamental differences between wireless signals and images or texts remain unexplained.
> > >
> > > We extended our existing discussion in Section 1 (text highlighted in red). In short, we highlight that unlike typically studied modalities (e.g., images, text), our task of predicting wireless signals is an inherently geometric problem that requires keeping track of many interactions of the signal with the environment.
> > >
> > > &nbsp;
> > >
> > > > However, before some experiment results are provided, I am not very convinced by the explanation for the Adam optimizer
> > >
> > > To provide empirical evidence, we trained GATr and Transformer with different optimizers on the DICHASUS data for 5k Tx positions (Section 5.4). After 50k steps, we evaluate the test performance. We obtain the following results:
> > > |  |   Adam |   RMSProp |   SGD |   SGD + Momentum |
> > > |:------------|-------:|----------:|------:|-----------------:|
> > > | GATr        |   0.68 |      0.67 |  0.75 |             0.71 |
> > > | Transformer |   0.69 |      1.4  |  0.92 |             0.74 |
> > >
> > > You can see here that Adam performs more robustly compared to the other optimizers for both models. Please note that we are not claiming that Adam is the best optimizer for this problem.
> > >
> > > Adam was also used in prior works relevant to the wireless domain (Geometric DL [1, 2, 3] and Wireless surrogates [4, 5, 6]).
> > >
> > > &nbsp;
> > >
> > > [1] Johannes Brandstetter, Rob Hesselink, Elise van der Pol, Erik J Bekkers, and Max Welling. Geometric and physical quantities improve E(3) equivariant message passing. In ICLR, 2022b.
> > >
> > > [2] Johann Brehmer, Pim de Haan, S¨onke Behrends, and Taco Cohen. Geometric Algebra Transformer.
> > > In NeurIPS, 2023.
> > >
> > > [3] Julian Suk, Pim De Haan, Baris Imre, and Jelmer M.Wolterink. Geometric algebra transformers for large 3d meshes via cross-attention. In ICML 2024 Workshop on Geometry-grounded Representation Learning and Generative Modeling, 2024.
> > >
> > > [4] Jakob Hoydis, Faycal Aıt Aoudia, Sebastian Cammerer, Florian Euchner, Merlin Nimier-David, Stephan ten Brink, and Alexander Keller. Learning radio environments by differentiable ray tracing. arXiv:2311.18558v1, 2023.
> > >
> > > [5] Tribhuvanesh Orekondy, Pratik Kumar, Shreya Kadambi, Hao Ye, Joseph Soriaga, and Arash Behboodi. Winert: Towards neural ray tracing for wireless channel modelling and differentiable simulations. In ICLR, 2022b.
> > >
> > > [6] Thomas M. Hehn, Tribhuvanesh Orekondy, Ori Shental, Arash Behboodi, Juan Bucheli, Akash Doshi, June Namgoong, Taesang Yoo, Ashwin Sampath, and Joseph B Soriaga. Transformer-based neural surrogate for link-level path loss prediction from variable-sized maps. In IEEE Globecom, 2023.

---

> ### Comment · Reviewer_HmDB · 2024-12-01
>
> Thanks for the clarification. I am updating the score to 8.

---

### Official Review · Reviewer_gkSN · 2024-10-31

**Soundness:** 3
**Presentation:** 2
**Contribution:** 2
**Rating:** 6
**Confidence:** 3

**Summary:**

The paper introduces Geometric Algebra Transformer (GATr) into wireless channel observation problem and builds a learnable neural simulation surrogate Wi-GATr to predict channel states based on scene primitives. The authors design a Wi-GATr Backbone to exploit the inherent geometric nature of the propagation of wireless signals. Further, they apply this model to probabilistic inference and receiver localization problems. Experimental results show that Wi-GATr outperforms other methods on the two datasets they constructed.

**Strengths:**

1. They design a new Wireless Geometric Algebra Transformer (Wi-GATr) backbone, which embeds the information of the wireless scene into geometric algebra while the network learns to model the channel.
2. They develop a learnable forward-model for channel simulation and an inverse-model for receiver localization based on the differentiable properties.
3. They build two new datasets with diverse scene geometry.

**Weaknesses:**

1. The problem of this work is not well identified. The authors only give the formulation of geometric algebra, but do not give any introduction of the wireless channel model. Wireless channels are complex and consist of many parameters. Authors need to specify what information about the channel they want to simulate and predict.
2. The challenges that need to be addressed are not clearly stated. The authors introduce GATr into this work and build a backbone to make it fit for the wireless channel prediction problem. However, the difficulties and challenges of model transfer are not fully introduced.
3. The innovation is somewhat limited. In addition to the designed backbone, the rest of the work consists only of two application experiments using the properties of the existing model.
4. The explanation for some of the pictures is inadequate. For example, Figure 1 shows the geometric surrogates for modeling wireless signal propagation. However, there is not enough explanation of this figure in the paper. It's hard to get the main point of it.

**Questions:**

1. How were the two datasets generated? Were they extracted from other datasets or were they simulated themselves using other tools. Is this sufficient as one of the contributions of the paper?

---

> ### Author Response · Authors · 2024-11-18
> **Response to initial reviewer comments**
>
> Thank you for your review and comments! In the following we discuss them in more detail and refer to relevant revised sections of our paper.
>
>
> > The problem of this work is not well identified. The authors [...] do not give any introduction of the wireless channel model.
>
> Thank you for pointing this out.
> We have added a background paragraph on channel models and signal propagation to our revised paper.
> In our experiments, we show examples of predicting non-coherent received power, band-limited received power, and delay spread.
>
>
> > The challenges [of model transfer] that need to be addressed are not clearly stated.
>
> Applying transformers to modelling wireless signal propagation in 3D poses two challenges:
> 1. What is an adequate representation of the input data?
> 2. How can a transformer generalize to novel coordinate systems?
>
> To address the first challenge, we show a comparison of a naive mesh representation to our tokenization scheme in Figure 3.
> To address the second challenge, we identified the symmetries of the problem and an adequate architecture, i.e. GATr.
> In order to apply GATr, one has to properly embed the mesh representation into the correct Geometric Algebra types (see Section 3.2).
> In our experiments, we show the significant impact using an equivariant architecture has on the problem, outperforming several baselines.
>
>
> > ... the work consists only of two application experiments using the properties of the existing model.
>
> It is true that we do not propose an entirely new architecture or algorithm in this work.
> However, we do show an entirely novel geometric deep learning approach to wireless simulation neural surrogates.
> We do think that our performant backbone with its new geometric tokenization scheme, a new approach to wireless inverse problems based on diffusion models and conditional sampling, the demonstration on several experiments, and the release of two new datasets to the community present enough useful innovation and insights at the intersection of wireless applications and geometric deep learning research.
> Beyond the technical contributions, we aim to impact both fields by highlighting the geometric nature of wireless channel modelling which represents a novel angle to this machine learning problem.
>
>
> > ... Figure 1 shows the geometric surrogates for modeling wireless signal propagation. However, there is not enough explanation of this figure in the paper...
>
> Thank you for bringing this to our attention! We have revised the caption of Figure 1 to improve clarity.
>
>
> > How were the two datasets generated? Were they extracted from other datasets or were they simulated themselves using other tools. Is this sufficient as one of the contributions of the paper?
>
> The datasets were simulated using state-of-the-art wireless ray tracing software (Remcom Wireless InSite as noted by other reviewer).
> The diverse scenes and detailed wireless simulation makes the dataset compelling for data-driven channel modeling and the geometric deep learning community.
>
> Edit: Fixed typo.

---

> > ### Comment · Reviewer_gkSN · 2024-11-25
> >
> > The author answered my question to some extent. I would change the rating to 6 marginally above the acceptance threshold.
> > However, the following questions remain:
> > 1. The added description of the wireless signal propagation part is not highly related to the latter wireless simulation model setup.
> > 2. The challenges that need to be addressed should be reflected clearly in the paper not only in the rebuttal. It's better to use a small paragraph to state this problem.
> > It's suggested that the manuscript should be polished further.

---

> > > ### Author Response · Authors · 2024-11-26
> > >
> > > Thank you for raising your score! We have added more precise references in line 191 and line 480 to provide more details what we are predicting in our experiments. Furthermore, we have rewritten and restructured the introduction of the challenges of 3D surrogate modeling (highlighted in red). Your feedback is immensely valuable. Thus, please let us know if you are still missing essential details in the description of the wireless channel.
> > >
> > > One additional clarification: In the background paragraph that we added in our first revision, we aimed to provide a high-level introduction of wireless channel modelling. We introduced the wireless channel from a geometric optics perspective. We believe this perspective enables the machine learning community to quickly and intuitively understand the parameters involved (power, phase, delay, polarization, material-dependence). In the paragraph, we refer to the Tse & Viswanath for a more detailed introduction to wireless channel modeling.

---

### Official Review · Reviewer_z8wX · 2024-11-03

**Soundness:** 3
**Presentation:** 3
**Contribution:** 3
**Rating:** 6
**Confidence:** 4

**Summary:**

In this paper, the authors introduce Wi-GATr, a fully learnable neural simulation surrogate designed to predict wireless channels based on indoor scene elements, including surface mesh, antenna position, and orientation. They employ an equivariant Geometric Algebra Transformer with a tokenizer for wireless simulation. The proposed method is validated using two distinct simulated datasets.

**Strengths:**

(1) Wireless channel prediction is essential in wireless systems, and developing a fully learnable neural simulation surrogate for predicting wireless channels is an emerging topic.

(2) The techniques and experimental results in this paper are solid. The authors apply their proposed model not only to channel prediction but also to two inverse problems: receiver localization and scenario generation. Various simulation results further validate the effectiveness of the proposed method.

(3) The authors have developed two new 3D wireless datasets to validate their model, which would be valuable resources for the wireless research community if published.

**Weaknesses:**

(1) The novelty of this paper in the machine learning component is unclear.

(2) The definitions of inverse problems lack clarity.

(3) This paper lacks of comparisons with public datasets for channel prediction and other state-of-the-art channel prediction models with NeRF and diffusion models.

**Questions:**

(1) The novelty in the machine learning aspect of this paper is unclear. It appears that the work mainly leverages the equivariant Geometric Algebra Transformer for channel prediction. The authors should clarify which components in the machine learning section present new contributions.

(2) The definitions of inverse problems lack clarity. For instance, the authors should provide a more detailed discussion and formulation for receiver localization. Additionally, the explanations of probabilistic inference with diffusion models on Page 5 are inconsistent with the discussion of diffusion models on Page 13, as the diffusion models used do not seem to follow the standard DDPM framework. The authors should include the training and sampling algorithms for the diffusion model utilized, as well as discuss the model's input.

(3) It would be beneficial to compare the proposed method on public datasets and with other models related to channel prediction, such as NeRF and diffusion models.

(4) Regarding generalization, the authors primarily validate their approach on two different datasets. It would be helpful to consider cross-dataset scenarios to assess performance in unseen conditions. Additionally, the authors should discuss the impact of parameters, such as varying simulated frequencies and the number of paths, on prediction performance.

(5) The authors should proofread the paper to correct typographical errors, such as “The The Tx and Rx locations are sampled uniformly within the bounds of the floor layouts.”

---

> ### Author Response · Authors · 2024-11-18
> **Response to initial reviewer comments**
>
> Thank you for your review and comments! In the following we discuss them in more detail and refer to relevant revised sections of our paper.
>
>
> > The definitions of inverse problems lack clarity. For instance, the authors should provide a more detailed discussion and formulation for receiver localization. Additionally, the explanations of probabilistic inference with diffusion models on Page 5 are inconsistent with the discussion of diffusion models on Page 13...
>
> Thank you for bringing this to our attention. In our revised paper, we have incorporated additional details on conditional sampling and receiver localization. Furthermore, we have extended the appendix with detailed descriptions of our masked training procedure, including pseudocode algorithms for training and sampling from our diffusion model. Overall, we use the DDPM formulation for the noise scheduler during training and we have adapted the notation in the appendix to clarify this.
>
>
> > This paper lacks of comparisons with public datasets for channel prediction and other state-of-the-art channel prediction models with NeRF and diffusion models.
>
> We would like to highlight that the DICHASUS dataset is publicly available and contains real-world measurements.
> Could you provide more concrete pointers which aspects of other public datasets you are missing, please?
>
>
> > The novelty in the machine learning aspect of this paper is unclear. It appears that the work mainly leverages the equivariant Geometric Algebra Transformer for channel prediction.
>
> You're right: the Geometric Algebra Transformer architecture is indeed a crucial component to our work.
> However, the architecture alone is not enough to solve practical wireless problems.
> We add both a novel tokenization scheme that allows us to represent 3D wireless problems in geometric algebra representations as well as inference algorithms, including a new approach to wireless inverse problems based on diffusion models.
> For practitioners, our release of two new datasets could be equally important.
> Beyond the technical contributions, we view our treatment of wireless channel modelling as a geometric deep learning problem as our novel key insight.
>
>
> > Regarding generalization, the authors primarily validate their approach on two different datasets. It would be helpful to consider cross-dataset scenarios to assess performance in unseen conditions. Additionally, the authors should discuss the impact of parameters, such as varying simulated frequencies and the number of paths, on prediction performance.
>
> By pretraining on simulated data and fine-tuning on real-world data, we have shown cross-dataset benefits of our model and our generated dataset.
> Generalization with respect to other frequencies represents an interesting, but separate, orthogonal problem to the one that we tackled, namely generalization across geometries.
>
>
> > ... correct typographical errors, such as “The The Tx and Rx locations are sampled uniformly within the bounds of the floor layouts.”
>
> Thank you for pointing out this typo in our appendix. We have addressed it in our revision.

---

> > ### Comment · Reviewer_z8wX · 2024-11-29
> >
> > The author partially addressed my questions. I'll maintain my current rating. The ratting 6 is marginally above the acceptance threshold.

---

### Official Review · Reviewer_ToCc · 2024-11-03

**Soundness:** 3
**Presentation:** 2
**Contribution:** 3
**Rating:** 8
**Confidence:** 2

**Summary:**

The paper presents a learnable approach to tackle the problem of indoor wireless simulation. The proposed architecture is based on a Geometric Algebra Transformer, and a new tokenizer is introduced, allowing to leverage a 3D representation of the scene by taking 3D primitives as input. The model can also be integrated into an inverse problem framework based on diffusion, allowing to retrieve the position of the transmitter, the receiver or the geometry of the scene. Two new datasets for wireless simulation are also presented. Experiments are conducted in synthetic and real settings.

**Strengths:**

- The presentation is clear and the paper is well-written. The problem at hand and coresponding challenges are well introduced to the reader.
- The quantitative and qualitative results show the superiority of the method in multiple settings, and with regards to multiple variables (number of training samples, number of training rooms / transmitters, etc..
- The versatility of the model is underlined by its adaptation to the inverse problem setting.

**Weaknesses:**

- Although the results on synthetic data are convincing regarding the contribution of the proposed architecture, the impact of the proposed architecture w.r.t. the transformer is not so clear on real data, although the authors explain this by the simplicity of the scene.
- The most competitive baseline (SEGNN) is not evaluated on the WiPTR dataset.

**Questions:**

- Why does data augmentation lead to poorer results in some cases in table 2 for the transformer baseline ?
- Are the input coordinates of the transmitter/receiver 2D or 3D for the proposed model ?
- For Rx interpolation in in-distributions experiments (l. ~329) in table 1, have the floor layouts been seen by the model during training ? If so, this should be explained more clearly, and why this setting is relevant.

Minor remarks:
- l. 297: |ap|^2 -> ap^2 ?
- l. 315: while -> While
- l. 789: The The -> The

---

> ### Author Response · Authors · 2024-11-18
> **Response to initial reviewer comments**
>
> Thank you for your review and comments! In the following we discuss them in more detail and refer to relevant revised sections of our paper.
>
>
> > The most competitive baseline (SEGNN) is not evaluated on the WiPTR dataset.
>
> We have implicitly mentioned in line 308 that training SEGNN runs out of memory on WiPTR.
> We have made this more explicit in our revision.
> Thank you for pointing this out!
> It is one of the advantages of our Wi-GATr approach that thanks to the transformer architecture, it scales much better to larger scenes than SEGNN.
>
>
> > Are the input coordinates of the transmitter/receiver 2D or 3D for the proposed model ?
>
> We consider everything in 3D, including the input coordinates of the transmitter and receiver.
> In the synthetic datasets, the receiver and transmitter positions are randomly sampled in 3D, making it essential to take the height into account.
> This is one of the novelties of our approach.
>
>
> > Although the results on synthetic data are convincing regarding the contribution of the proposed architecture, the impact of the proposed architecture w.r.t. the transformer is not so clear on real data, although the authors explain this by the simplicity of the scene.
>
> We argue that the main strength of our architecture is the generalization capabilities to new geometric layouts.
> Currently, real-world datasets are lacking variety in geometric layouts, collecting data on multiple scenes is challenging and time-consuming.
> Therefore, we show the generalization capabilities on simulated data.
> On the single real-world scene, where the geometry remains fixed throughout, the transformer performs competitively.
> However, as we evaluate on unseen scenarios, such as rotated or translated scenes (Table 1), Wi-GATr remains competitive and robust, while a baseline transformer fails.
>
>
> > Why does data augmentation lead to poorer results in some cases in table 2 for the transformer baseline ?
>
> This is indeed surprising.
> We explain these results with the fact that the samples in the dataset have a lot of accidental shared structure, for instance that floor and ceiling are parallel to the x-y plane.
> E(3) augmentations remove this structure, so the network has to learn more from data.
> Given enough capacity and training steps, this should not be a problem, but within the fixed settings of our study this turned out to be more difficult to pick up for the transformer.
>
>
> > For Rx interpolation in in-distributions experiments (l. ~329) in table 1, have the floor layouts been seen by the model during training ? If so, this should be explained more clearly, and why this setting is relevant.
>
> Yes, in the Rx interpolation setting, the floor layouts have been seen during training.
> However, receiver locations we evaluate on were not seen during training.
> This evaluation tests the generalization to unseen Rx positions, testing the capabilities as a wireless simulator.
> The real-world relevance is highlighted in Section 5.4, where the simulations tuned on sparse data are used to predict missing measurements.
>
> We have rephrased the relevant parts in our revised paper to improve clarity.
> Thank you!
>
>
> > Minor remarks:
> >
> > l. 297: |ap|^2 -> ap^2 ?
>
> Thank you for pointing out the two typos. We have fixed them in the revision.
> Note that |a_p|^2 is more precise as a_p denotes the complex path gain.
> We have added a paragraph with background on channel modelling in Section 2 to clarify the notation.

---

> ### Comment · Reviewer_ToCc · 2024-11-26
>
> I thank the authors for clarifying the points mentioned, I updated my score to 8.

---

> > ### Author Response · Authors · 2024-11-26
> >
> > We are happy that our response has clarified your open questions. Thank you for your feedback and for raising your score.

---

### Author Response · Authors · 2024-11-18

We thank all reviewers for their insightful comments and suggestions.
We are glad to read that the reviewers appreciate the relevance and introduction of the problem (z8wX,ToCc), the originality of our approach (HmDB), the versatility and performance of our results (z8wX, ToCc, HmDB, gkSN), and the value of our datasets for research community (z8wx, gkSN).
Based on their feedback, we were able to improve our paper as discussed in the individual responses.
Our changes in the revision are highlighted in red.


We are looking forward to an interesting and constructive discussion!

---

### Meta-Review · Area_Chair_vsq6 · 2024-12-19

**Metareview:**

In this paper, the authors present a novel geometric transformer model for wireless simulation. The reviewers are generally positive and recognize the valuable contribution of the proposed transformer model, particularly its application to wireless indoor modeling. This work has the potential to advance the deployment of indoor communication nodes, such as pico-cells in 5G networks.

The authors focus on indoor systems, but it would be valuable to include a discussion of outdoor systems in highly dense urban environments, where propagation simulators may encounter similar limitations. However, the proposed transformer model might also face challenges related to scalability in 3D spaces and over larger distances. Additionally, the paper would greatly benefit from providing links to the code and data, as replicating such a tool without these resources would be challenging.

**Additional Comments On Reviewer Discussion:**

The authors and reviewers engaged constructively during the discussion phase, and the authors successfully convinced the reviewers of the merits of their work.

While the paper is strong, it would be significantly improved by releasing the code and data, as this would enhance its reproducibility and impact. However, I am uncertain whether acceptance can be made conditional on providing these resources.

In my opinion, this paper is not as strong as the reviewers perceive it to be. I believe most communication engineers would likely prefer ray tracing algorithms, especially when the computational complexity of the proposed method does not offer a significant improvement (e.g., a 5x speed-up). Additionally, the proposed model may perform poorly in out-of-distribution (OOD) cases. However, since the reviewers were very positive about the work, I chose not to intervene in their evaluation process. While this is not a bad paper, I believe its applicability may be limited. Releasing the code and data would help enhance its practical utility and impact.

---

> ### Public Comment · ~Thomas_Hehn1 · 2025-02-28
>
> We thank the AC and the reviewers for the fruitful discussion and their decision.
>
> We want to highlight that it is not our intention to generally replace ray tracing algorithms. We showed our method can solve problems that state-of-the-art ray tracing cannot solve. We invite interested researchers to try this out themselves using our open-source code. The links to the repositories can be found in the camera-ready version of the paper and the code will be published there by the time of the conference.

---

### Decision · Program_Chairs · 2025-01-22

Accept (Poster)